# Spatially revealed roles for lncRNAs in *Drosophila* spermatogenesis, Y chromosome function and evolution

Zhantao Shao[1], Jack Hu[1], Allison Jandura[1,2,3], Ronit Wilk[1], Matthew Jachimowicz[1,2,3], Lingfeng Ma[1,2,3], Chun Hu[1], Abby Sundquist[1], Indrani Das [1], Phillip Samuel-Larbi [1], Julie A. Brill [2,3,4] ✉ & Henry M. Krause [1,2,4] ✉

Unlike coding genes, the number of lncRNA genes in organism genomes is relatively proportional to organism complexity. From plants to humans, the tissues with highest numbers and levels of lncRNA gene expression are the male reproductive organs. To learn why, we initiated a genome-wide analysis of *Drosophila* lncRNA spatial expression patterns in these tissues. The numbers of genes and levels of expression observed greatly exceed those previously reported, due largely to a preponderance of non-polyadenylated transcripts. In stark contrast to coding genes, the highest numbers of lncRNAs expressed are in post-meiotic spermatids. Correlations between expression levels, localization and previously performed genetic analyses indicate high levels of function and requirement. More focused analyses indicate that lncRNAs play major roles in evolution by controlling transposable element activities, Y chromosome gene expression and sperm construction. A new type of lncRNA-based particle found in seminal fluid may also contribute to reproductive outcomes.

Long noncoding RNAs (lncRNAs) have been simply defined as non-coding RNAs longer than 200 nucleotides in length. Given these simple criteria, it is not surprising that lncRNAs carry out a wide range of complex functions (reviewed in ref. 1). In metazoans, lncRNA genes comprise a large portion of the genome, with numbers correlating roughly to morphological and neuronal complexity[2]. In mammals, lncRNA gene numbers exceed those of coding genes by an order of magnitude. Early studies suggested that the majority of these genes are either poorly or not expressed in most cell types[3], casting doubt on their overall importance and function. These conclusions were based on RNAseq and scRNAseq analyses that made use of poly A selection during RNA isolation and detection, which we show here excludes a large proportion of lncRNAs.

Unlike past large-scale lncRNA gene analyses, the examination of lncRNAs here by in situ hybridization does not depend on RNA 3' end

status and provides spatial and subcellular information that lends considerable insight into potential functions. We initiated this study using high-resolution fluorescent in situ hybridization (FISH) to analyze lncRNA cellular and subcellular expression patterns in *Drosophila* male reproductive glands, where the largest numbers of lncRNAs are expressed[4–7].

The production of male gametes represents a unique phase of development with extremely high genome accessibility[8–10]. Consequently, from plants to animals, male gametogenesis is where the highest numbers of genes are expressed[11–14]. Notably, the Y chromosome, normally heterochromatic and silenced in all other tissues, is also rendered highly accessible and broadly transcribed during spermatogenesis. Within this temporarily accessible genomic landscape are vast arrays of repetitive elements that range from numerous types of transposons to expansive satellite repeats. Although this broad

[1]Donnelly Ctr., 160 College St., University of Toronto, Toronto, ON, Canada. [2]Department of Molecular Genetics, University of Toronto, Toronto, ON, Canada. [3]Cell Biology Program, The Hospital for Sick Children, Toronto, ON, Canada. [4]These authors jointly supervised this work: Julie A. Brill, Henry M. Krause. ✉e-mail: julie.brill@sickkids.ca; h.krause@utoronto.ca

exposure of repetitive elements creates the potential for deleterious expression and genomic rearrangements, it also presents opportunities for beneficial new gene production. Indeed, most new gene evolution in complex multicellular organisms appears to occur during gametogenesis (reviewed in refs. 4–6,14). Our analyses here of lncRNA subcellular distributions and content, together with links to functional analyses, suggest high levels of functionality, with many of these functions related to repetitive element management, Y chromosome activities, sperm production and various types of paternal contributions.

## Results

The *Drosophila* male reproductive tract contains four major organs (Fig. 1a), all of which were included in this study. These include testes, where sperm are generated, seminal vesicles, where sperm are stored, accessory glands, which produce seminal fluid, and the ejaculatory duct, where sperm and seminal fluid are mixed prior to transfer. A more detailed diagram of a testis (Fig. 1b) shows the main concurrent stages of spermatogenesis, with germline and somatic stem cells positioned at the anterior tip, and derived gonialblasts, spermatogonia, spermatocytes, spermatids, and sperm forming in succession. Fig. 1c shows an actual testis examined by FISH for the lncRNA *CR44792* expression. Expression of lncRNAs was observed in all cell types of all four tissues, with percentages of lncRNAs detected and subcellularly localized indicated in Fig. 1d. Testis expression patterns were observed for 88% of the of >600 lncRNAs examined thus far, with accessory glands a close second at ~81% of lncRNAs tested. In testes, the cell types that express the highest numbers and levels of lncRNAs are spermatocytes (SCs) at 61% and spermatids (SPs) at 82% (Fig. 1e). The spermatid number is extremely interesting, as the number of coding genes transcribed during this time is extremely low (<0.5%[15]). The majority of lncRNAs detected in spermatids must require post-meiotic transcription, as expression levels during earlier stages are much lower or absent in more than 75% of cases. Notably, in the prior RNAseq analysis[13], expression levels of most detected lncRNAs were annotated as "low", "very low" or "not detected", whereas expression detected here by FISH is much more robust (addressed further below).

As we have found previously in other tissues[16], subcellular lncRNA trafficking distributions are frequent and diverse, with a high of ~81% of expressed lncRNAs clearly subcellularly localized in accessory glands (Fig. 1d). With all four tissues considered, 87% of expressed lncRNAs show clear subcellular localization in at least one tissue. Surprisingly, in contrast to previous speculation that lncRNAs are primarily found in the nucleus[3,17], less than 15% of those tested exhibit nuclear patterns (Fig. 1f), and many of those that do also exhibit cytoplasmic distributions, either simultaneously or in a different stage or cell type.

The varieties of cellular and subcellular patterns of testis lncRNA expression observed are both numerous and diverse. Examples for early and late germline stages, as well as cyst cell patterns, are provided in Supplemental Figs. 1–3. These include patterns from all stages of spermatogenesis, from stem cells to mature sperm, suggesting roles in virtually all aspects of spermatogenesis and subsequent sperm function. Examples of expression patterns for the other three tissues are shown in Supplemental Fig. 4. All of the images obtained, along with annotations, are available on our *Drosophila* RNA expression database "Fly-FISH" (http://fly-fish.ccbr.utoronto.ca). This database is searchable by gene name or localization pattern, making it useful for finding genes with common or exclusionary patterns of cellular and subcellular expression. The former often implies genes with related gene functions.

### LncRNA polyadenylation

As noted above, a large proportion of the lncRNAs that we examined show robust levels of expression, in contrast to levels determined by previous RNAseq[13] and scSEQ[7] studies. We have suggested previously[6,18,19] that differences in lncRNA detection levels using FISH versus RNAseq may be due, in part, to the use of poly A-containing RNA selection (many lncRNAs have been shown to lack poly A tails[20,21]). To further investigate this possibility, we searched for poly A signals (PASs) in the transcripts of all *Drosophila* coding and lncRNA genes. In line with previous studies[22], we found the consensus PAS, AATAAA, in 60% of protein-coding genes, but only 23% of lncRNA genes (Fig. 1g). While degenerate PAS sequences make up for lack of the consensus PAS in many coding genes, the prevalence of these is also lower in lncRNA genes, and less likely to occur in tandem with others or with a downstream signal sequence (DSS; Supplementary Fig. 5). Thus, we expect that significantly more than 50% of lncRNAs may not be polyadenylated or are only partially polyadenylated.

To explore this further, we looked for PAS sequences in lncRNA genes that were deemed highly expressed by FISH, RNAseq and scRNAseq versus those found to be abundant via FISH analysis but very low or not detected by RNAseq/scRNASeq[7,13]. All genes in the first category have strong consensus PAS motifs, while 80% of those in the latter category do not. Further analysis of a subset of these by RT-PCR confirmed the suspected presence or absence of poly A tails (Fig. 1h). All of the lncRNAs tested that were readily detected by FISH and not RNAseq are not reverse-transcribed by oligo dT primers (lanes 7–12 top panel) but show robust reverse transcription and amplification when using random or specific 3' primers (bottom panel). Notably, among the set of lncRNAs recently found via CRISPR-based analyses to be required for male fertility[23], less than 25% contain a PAS. Thus, polyadenylation appears not to be present or required for a large proportion of lncRNAs.

### Correlations between expression and function

Studies on mRNA subcellular trafficking have shown that most mRNAs are subcellularly localized prior to translation near sites of protein requirement[16,24]. To gain insight into whether subcellular localization is also important for lncRNA functions, we looked for potential correlations between subcellular localization and genetic requirement. Looking at 99 of the 105 testes-expressed lncRNAs recently analyzed by CRISPR analyses[23], we observed strong correlations between requirements for fertility and both levels of expression (Fig. 1i) and subcellular localization (Fig. 1j). Thus, the overall high levels of expression and subcellular localization found here suggest similar overall levels of lncRNA functionality. Previous analyses of lncRNA gene sequence conservation have also suggested relatively high levels of lncRNA gene function[6,23]. To delve deeper into some of these potential functions, we selected a number of localization categories and genes that might explain the propensity and need for lncRNA expression in these tissues.

### Y-loop expression patterns

Twenty-three of the initially examined lncRNAs (Table 1) are found within spermatocyte nuclei in patterns that resemble previously described "Y-loops" (e.g., *CR43622*, Fig. 2a). Y-loop RNA patterns are formed by nascent transcripts produced by extremely large Y chromosome coding genes whose DNA sequences extend away from most of the peripherally located Y chromosome, deep into the nucleoplasm[25,26]. The enormous sizes of these genes (up to ~4 million base pairs) is due to expansive introns composed primarily of satellite repeats and transposable elements, some of which have still not been fully sequenced and annotated[27]. Double-labeling with *CR43622* and an antibody that marks Y-loop structures[28] reveals extensive overlap (Fig. 2b). This co-localization suggests possible roles for these lncRNAs in Y-loop "mega-gene" transcription, stability, packaging or processing. Notably, eight of the Y-loop localizing lncRNAs are among the 105 lncRNAs previously tested for function[23], with five of these causing moderate to severe infertility when knocked out (checkmarks vs. X's in

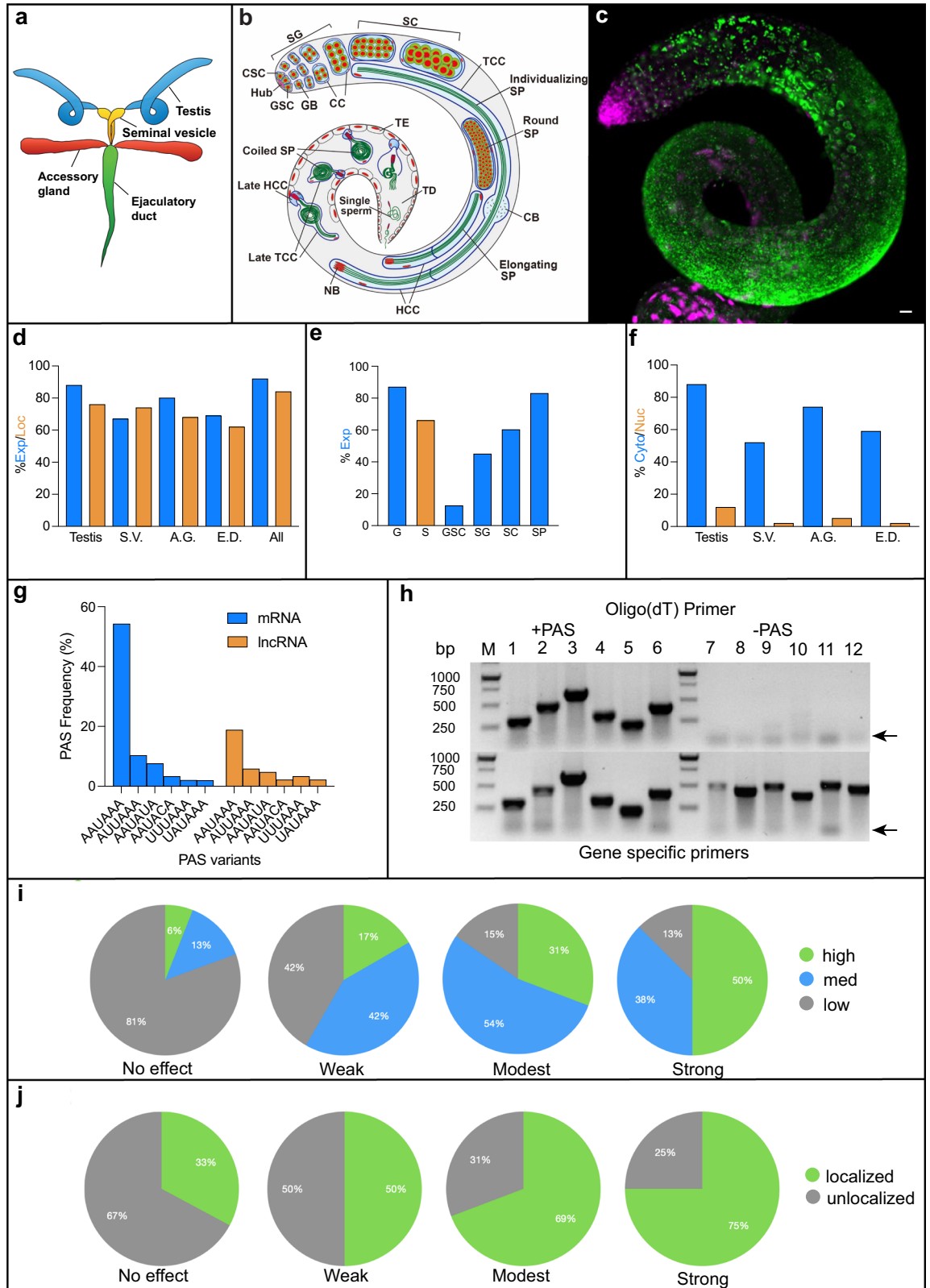

Table 1). Those with decreased fertility exhibited sperm differentiation and morphology defects similar to those exhibited by mega-gene mutations[23], consistent with roles in Y-loop gene expression or processing. Consistent with these needs, we note that all five of the required lncRNAs are conserved in other *Drosophila* species, whereas two of the three non-essential lncRNAs are not. The latter may represent newly evolved or redundantly acting lncRNA genes.

In our previous[16,24] and ongoing work, we have noted a strong tendency for genes located near one another on chromosomes to be co-regulated and functionally related. To see if Y-loop-expressed lncRNA genes might reside near other genes with potential Y-loop production or processing functions, we examined their neighboring genes. We first note that two of the Y-loop lncRNA genes on our list reside next to one another (*CR43431*, *CR44553*; Table 1). Several other

**Fig. 1 | Male reproductive tissues and lncRNA metrics. a** Schematic of reproductive tissues. **b** Schematic diagram of testis. Spermatogenesis begins at top left with the stems cells and ends where mature sperm exit the terminal duct (TD). CSC cyst stem cell, GSC germline stem cell, SG spermatogonium, SC spermatocyte, SP spermatid, CB cystic bulge, TCC tail cyst cell, HCC head cyst cell, GB gonialblast, NB nuclear bundle, TE terminal epithelium. **c** Example of testis expression of lncRNA *CR44792*. RNA is green, DAPI is magenta. Scale bar = 20 μm. **d** Percentages of tested lncRNAs expressed (blue) and subcellularly localized (orange) in each of the four assessed tissues. **e** Percentages of lncRNAs expressed in germline cells (G: blue) and somatic cells (S: orange). Abbreviations are as described in (**b**). **f** Percentages of lncRNAs found in the cytoplasm (blue) or nucleus (orange). S.V. seminal vesicle, A.G. accessory gland, E.D. ejaculatory duct. **g** Percentages of mRNAs (blue) and lncRNAs (orange) with indicated polyadenylation signals (PAS). When multiple PAS signals were detected, only the closest to consensus was counted. Frequencies

showing additional PAS motifs and downstream sequence elements are shown in Supplemental Fig. 1. **h** RT-PCR results for lncRNAs containing (+PAS) or missing (-PAS) PAS motifs. Genes in the top panels used 3′ end oligo dT primers while those on the bottom used gene-specific 3′ end primers. Arrows indicate non-specific primer-dimer products. lncRNA gene identities (1–12), 3′ end sequences and reverse primers are provided in Supplemental Table 1. **i** Correlations between relative levels of testis lncRNA expression (based on exposure time) and requirements for fertility[23,62]. Fertility effects were divided into four groups: unaffected, weakly affected, moderately affected and strongly affected. Expression levels were divided into three groups, low (gray), medium (blue) and high (green). **j** Correlations between lncRNA subcellular localization in testes and effects on fertility (as defined in panel i). Relative percentages of non-localized (gray) or subcellularly localized (green) lncRNAs for each fertility group. Original images and numbers are available in the provided source data.

## Table 1 | List and characteristics of Y-loop localizing lncRNAs

| lncRNA | Chromosome | Gene size | Intron | Proximal genes | Poly-A | Conserved |
|---|---|---|---|---|---|---|
| CR42657 | X | 1 kb | 0.2 kb, 64 bp | CR44016, *nej*, *btd* | No | No |
| CR43193 | X | 1 kb | 50 bp | Shakb* | Yes | Yes |
| **CR43431** | 3L | 5 kb | 2 kb, 54 bp, 67 bp | CR32111, CR44552, **CR44553** | No | Yes |
| CR43622ˣ | 2R | 0.6 kb | N | hbs | Yes | Yes |
| CR43835ˣ | X | 0.9 kb | 58 bp, 62 bp, 62 bp | CR43279, CG32719 | No | No |
| CR43839√ | 2L | 0.8 kb | N | CR45722, CR45723, *tj* | Yes | Yes |
| CR43862√ | X | 0.9 kb | 91 bp, 77 bp | CR45610, *run* | No | Yes |
| CR44017 | 3R | 0.6 kb | N | *gpp** | Yes | Yes |
| CR44114 | 2L | 0.9 kb | 58 bp | rotini | No | Yes |
| CR44206 | 2R | 15 kb | 7.2 kb, 6 kb | Dh31-R, CR45312 | No | Yes |
| CR44344√ | 2R | 2 kb | 0.4 kb, 57 bp, 0.2 kb | *mbl** | No | Yes |
| CR44412√ | 2L | 0.8 kb | N | *Sytα* | No | Yes |
| **CR44553** | 3L | 1 kb | N | **CR43431**, CR44552, CR32111 | No | Yes |
| CR44606 | 2L | 1 kb | N | sky*, CG42238* | Yes | No |
| CR44619 | X | 2.7 kb | 1.4 kb | lncRNA:Vinr, paics | Yes | No |
| CR44792√ | 2L | 1 kb | N | CG31475 | No | Yes |
| CR44889 | X | 1 kb | 56 bp | *meso18E*, CG12531 | No | Yes |
| CR44977 | 2L | 1.4 kb | 81 bp | *Brat** | No | Yes |
| CR45174ˣ | 3L | 1 kb | N | *Ago3** | No | No |
| CR46499 | X | 9.9 kb | 1.3 kb, 1.4 kb + 9 more | CR43614, CR32773*, *NorpA** | No | Yes |
| CR45805 | 3L | 3.3 kb | N | Pdk1*, CR45400, 45401, 45402 | Yes | Yes |
| CR45965 | 3L | 1.4 kb | N | Ten-m*, CR45962*, 45964, 45963 | No | Yes |
| CR46278 | 2R | 2 kb | N | *lola**, psq* | No | No |
| CR40441 | Y | 0.2 kb | N | Ppr-Y* | No | No |
| CR45931 | Y | 0.6 kb | N | WDY* | No | Yes |
| CR45946 | Y | 0.5 kb | N | kl-3* | No | Yes |
| CR45947 | Y | 0.5 kb | N | kl-3* | No | Yes |
| CR45948 | Y | 0.9 kb | 61 bp | WDY* | No | Yes |

Columns from left to right indicate lncRNA annotation numbers, chromosomal location, gene size, presence and size of introns, proximal genes, presence of a consensus Poly A signal, and sequence conservation between *Drosophila* species. Check marks adjacent to lncRNA annotation symbols indicate moderate to severe infertility when knocked out, whereas X's indicate no or minor phenotypes. Y-loop lncRNA genes found proximal to each other are in bold. lncRNAs with no introns are labeled as "N". Proximal gene products involved in RNA transcription or processing are indicated in italics. Asterisks indicate genes with unusually long introns. The five Y chromosome-linked Y-loop lncRNAs discovered last are listed at the bottom of the table.

unusual features were found to be enriched among flanking or overlapping genes. First, many of the flanking genes, like the Y chromosome mega-genes, contain unusually large introns (Table 1; asterisks), raising the possibility that some Y-loop lncRNA genes may have general roles in the expression or processing of large gene transcripts. Interestingly, many of the proximal/overlapping gene products also have roles in RNA transcription or processing (e.g., *Ago3, mbl, Brat, meso18E, tj;* indicated in italics).

None of the lncRNA genes listed in Table 1 are located on the Y chromosome. Upon perusal however, we did find several annotated lncRNAs that are positioned within or near Y chromosome mega-

genes. Designing probes to test these, we found that, despite coming from different Y chromosome regions, several are nearly identical to one another in sequence, and occasionally to Y-loop gene exon sequences (Supplementary Fig. 6). For example, *CR45946* and *CR45947*, which are upstream of *kl-3* and within the *kl-3* first intron, respectively, share nearly identical sequences with each other and with *CR45931*, which is just upstream of the Y chromosome mega-gene *WDY*. All three are also nearly identical to the first ~500–600 nucleotides of the *WDY* first exon (Supplementary Fig. 6). Based on their positions and orientations, this provides all three lncRNAs, as well as the *WDY* first exon, with potential to hybridize to the first intron of *kl-3*, and/or potential

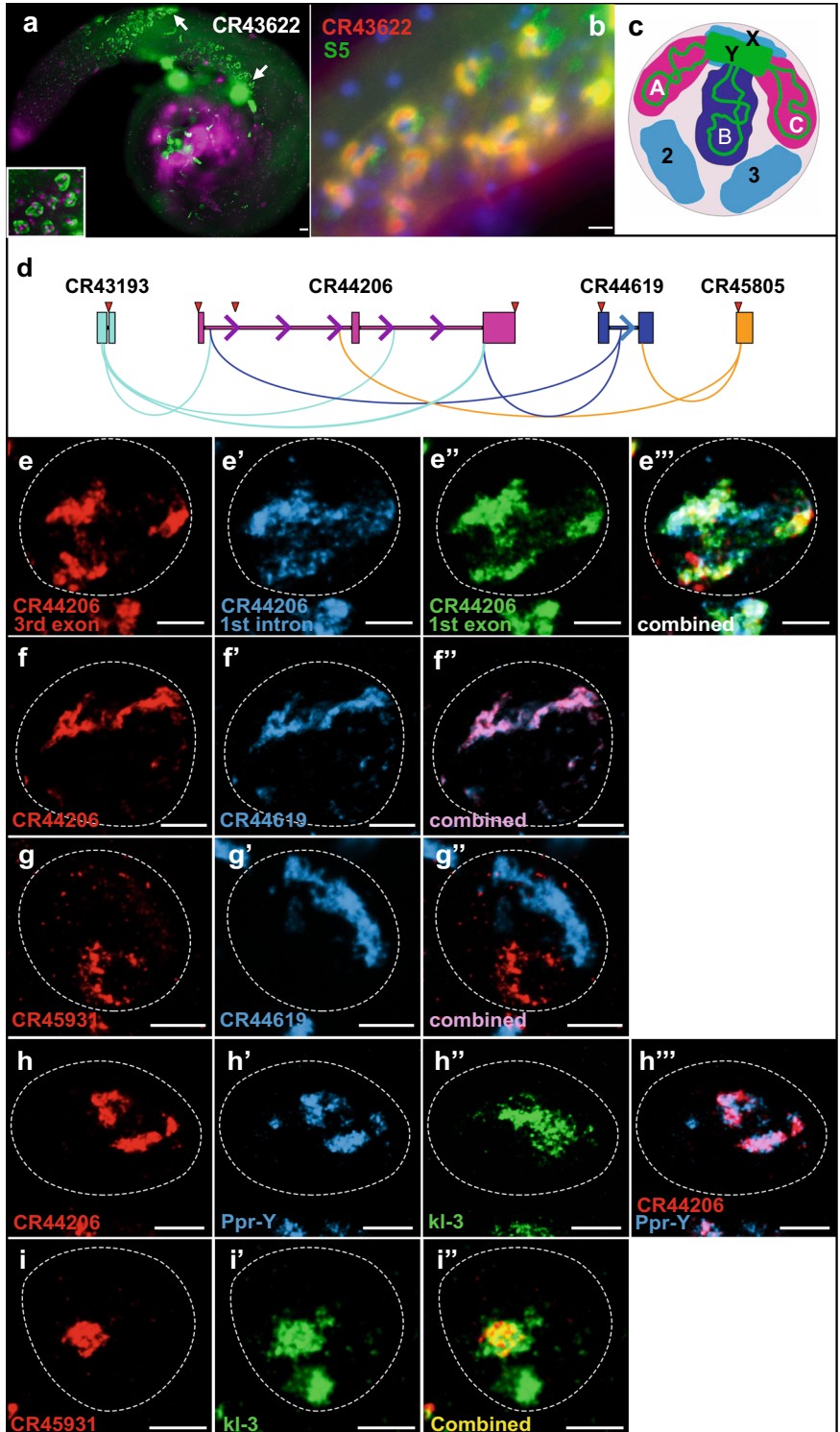

**Fig. 2 | LncRNA expression in Y-loops. a** Typical 'Y-loop' expression of *CR43622* (green) in spermatocyte nuclei (magenta). **b** Co-localization of *CR43622* (red) with the Y-loop-associated S5 protein (green). DAPI is blue. **a**, **b** Scale bars = 20 μm. **c** Schematic of a late spermatocyte nucleus (adapted from[25]) indicating approximate locations of chromosomes X, Y, 2, and 3. Relative positions of the A, B, and C loops produced by Y mega-genes *kl-5*, *kl-3*, and *ks-1/ORY* (green lines) and their transcripts (pink and dark blue shading). **d** Diagram of Y-loop lncRNA *CR44206* and three predicted lncRNA interactors. Exons (thick rectangles), introns (thin lines), transcription direction (arrows), probe-target sites (triangles). Complementary sites are illustrated with curved lines connecting complementary pairs. **e–i** HCR FISH using oligonucleotide probes that specifically target **e** *CR44206* 3rd exon (e), 1st intron (e'), 1st exon (e") and combined (e‴) in a single primary spermatocyte nucleus, **f** *CR44206* and predicted interactor *CR44619*, **g** *CR44619* and predicted non-interactor *CR45931*, **h** *CR44206* and two mega-gene transcripts, *Ppr-Y* and *kl-3* or **i** *CR45931* and the mega-gene *kl-3*. Individual spermatocyte nuclei are outlined in white dashed circles. **e–i** Scale bars = 5 μm. Images are projected confocal stacks with Z-step size 0.07 μm × 50 slices.

upstream regulatory elements of both coding genes. Alternatively, these sequences may allow the sharing of interacting RNA and/or protein cofactors. Two other mega-gene-associated lncRNAs are present in the last introns of *WDY (CR45948)* and *Ppr-Y (CR40441)*. Interestingly, *CR45948* is identical to the last *WDY* exon (3'UTR excluded), including a short 61nt intron. Using HCR FISH, small oligonucleotide probes that recognize each of these Y chromosome lncRNAs all produce Y-loop expression patterns. In total, after the detection and testing of additional lncRNA genes with potential Y-loop patterns and functions, 28 Y-loop-associated lncRNA transcripts were identified (Table 1).

Thinking that these lncRNAs may co-localize, in part, by hybridizing with complementary mega-gene sequences and/or one another, we conducted a genome-wide search for complementary or identical sequences for each of the 28 Y-loop lncRNA genes. In doing so, we found that the largest of the Y-loop co-localizing transcripts, *CR44206*, has numerous complementary sequences to three of the other Y-loop localizing transcripts (*CR43193, CR44619*, and *CR45805*). Interestingly, these potentially interacting sequences are all within the two *CR44206* introns (Fig. 2d). Since these introns are annotated as poorly expressed on FlyBase (http://flybase.org), along with the first exon, we generated short oligonucleotide probes that specifically target each exon or intron for detection by HCR FISH. Figure 2e shows that all probes yielded similar expression patterns, indicating that *CR44206* transcripts accumulate in Y-loops in unspliced form. Notably, CR44206 also lacks PAS motifs, which may explain the low expression levels noted on FlyBase.

Next, we looked to see if the predicted *CR44206*-interacting lncRNAs also co-localize. Figure 2f shows that, as predicted, *CR44206* and *CR44619* show highly similar Y-loop sub-expression patterns, as do the other two predicted interactors, *CR43193* and *CR45805* (Supplementary Fig. 7). In contrast, *CR45931*, a Y-loop localizing lncRNA not predicted to interact with *CR44206*, shows a non-overlapping Y-loop pattern (Fig. 2g), suggesting that it interacts with a different mega-gene transcript and cofactors.

*CR44206* and its three co-localizing lncRNAs also contain sequences that are complementary to the Y chromosome mega-gene *Ppr-Y*. To see if these lncRNAs also co-localize with *Ppr-Y*, we performed double labeling with probes that recognize *CR44206* and *Ppr-Y*. Figure 2h shows that both probe sets extensively co-localize to similar nuclear targets. In contrast, probes that recognize the mega-gene *kl-3* produce a non-overlapping pattern with *CR44206*, indicating that *Ppr-Y*, a gene that has not been previously tested for spatial expression, and *kl-3*, which has previously been shown to label the Y-loop "B" domain[25] (Fig. 2b), accumulate within independent structures. Double-labeling using probes that target the *CR44206* non-co-localizing lncRNA *CR45931* and the mega-gene *kl-3* show that these do label overlapping, though not identical, regions (Fig. 2i). Further efforts will be required to determine the full complexity of Y-loop gene nuclear expression subdomains and lncRNA interactions.

## Other transcribed repetitive elements

As previously noted, differentiating germline cells in testes have the highest levels of chromatin accessibility[8–10], which exposes repetitive elements such as the mega-gene introns and vast arrays of transposable elements (TEs). This provides the potential for disruptive TE expression, mobility, and subsequent chromosome rearrangements. Much of this activity is known to be suppressed by noncoding genes on the X and Y chromosomes. Two of these were investigated here.

The X chromosome noncoding gene *flamenco (flam)* has been shown to play a key role in TE element silencing during oogenesis[29–31]. The 220 kb gene is highly enriched for TEs and other repeated sequences within both its exons and expansive introns. Spliced *flam* transcripts are subsequently converted into piRNAs that can silence several types of retrotransposons. Our random selection of lncRNA

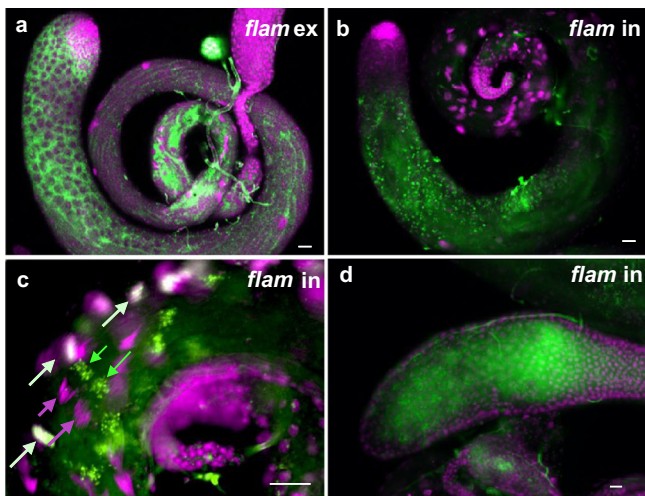

**Fig. 3 | Flamenco exon and intron expression patterns. a** *flamenco (flam)* expression detected in the testis using a 1st exon probe. Expression is cytoplasmic. **b** *flam* expression detected using an intronic probe. Expression is primarily nuclear (Y-loop-like in spermatocytes). **c** *flam* transcripts detected both on (white arrows) and trailing (green arrows) elongated spermatid nuclei (magenta). **d** *flam* transcripts in secretory cells and lumen of accessory glands. Scale bars = 20 μm.

cDNA-derived probes for FISH analysis included probes that recognize the *flam* first exon and intron, both of which give strong signals in spermatocytes and spermatids. The 1st exon probe yields cytoplasmic expression patterns, as expected for a transcript destined for piRNA production (Fig. 3a). In contrast, the 1st intron probe yields a pattern of nuclear foci (Fig. 3b). This is also true for oligonucleotide HCR FISH probes that target other parts of the large 1st intron (Fly-FISH). These nuclear transcripts may silence TE expression epigenetically. Figure 3c also shows an unusual late pattern for intronic sequences in elongating spermatids, with transcript foci decorating elongating nuclei early and then appearing to move distally later. Relatively high levels of expression are also seen in the accessory gland (Fig. 3d) and seminal vesicle lumens (Fly-FISH), raising the possibility that *flam* activity could be transferred paternally (see "Discussion").

The second set of repeat-regulating genes that we focused on was the Y-linked cluster of *Suppressor of Stellate (Su(Ste))* noncoding genes, which as implied by name, are required to repress expression of X-linked *Ste* gene clusters (reviewed in ref. 32). Both exist in multiple repeat copies (>400 for *Su(Ste)*, ~40 for *Ste*[27]), with *Ste* repeats primarily on the X chromosome. Both clusters evolved from a common protein-coding precursor, and share similar sequences over most of their lengths, but only the *Ste* genes remain protein-coding. In the absence of *Su(Ste)* genes, unsuppressed *Ste* expression results in the formation of Ste protein crystals, leading to male sterility[33,34]. Importantly, each of the *Su(Ste)* repeats is transcribed in both directions, with antisense transcripts (*Su(Ste)-AS)* under the control of *hoppel/1360* transposable elements inserted at the 3'ends of each *Su(Ste)* repeat unit (Fig. 4a). These antisense transcripts are processed into piRNAs that are required for *Ste* transcript degradation[35].

Several *Su(Ste)* cDNAs, both sense and antisense, were among the cDNAs used to generate our initial lncRNA probes. Although a previous in situ hybridization study reported that there is little transcription from *Su(Ste)* repeats in the sense direction[36], our results indicate that both transcripts are produced, beginning with nuclear accumulation in the early germline, and then cytoplasmic accumulation (Fig. 4b–d). These *Su(Ste)* transcripts may also play later roles in spermatogenesis, as both the sense and antisense transcripts reappear during spermatid stages of development (Fig. 4b; Fly-FISH). Like *flam*, these later patterns include foci on elongating spermatid nuclei and particles in the accessory gland lumen, suggesting that they may have paternal

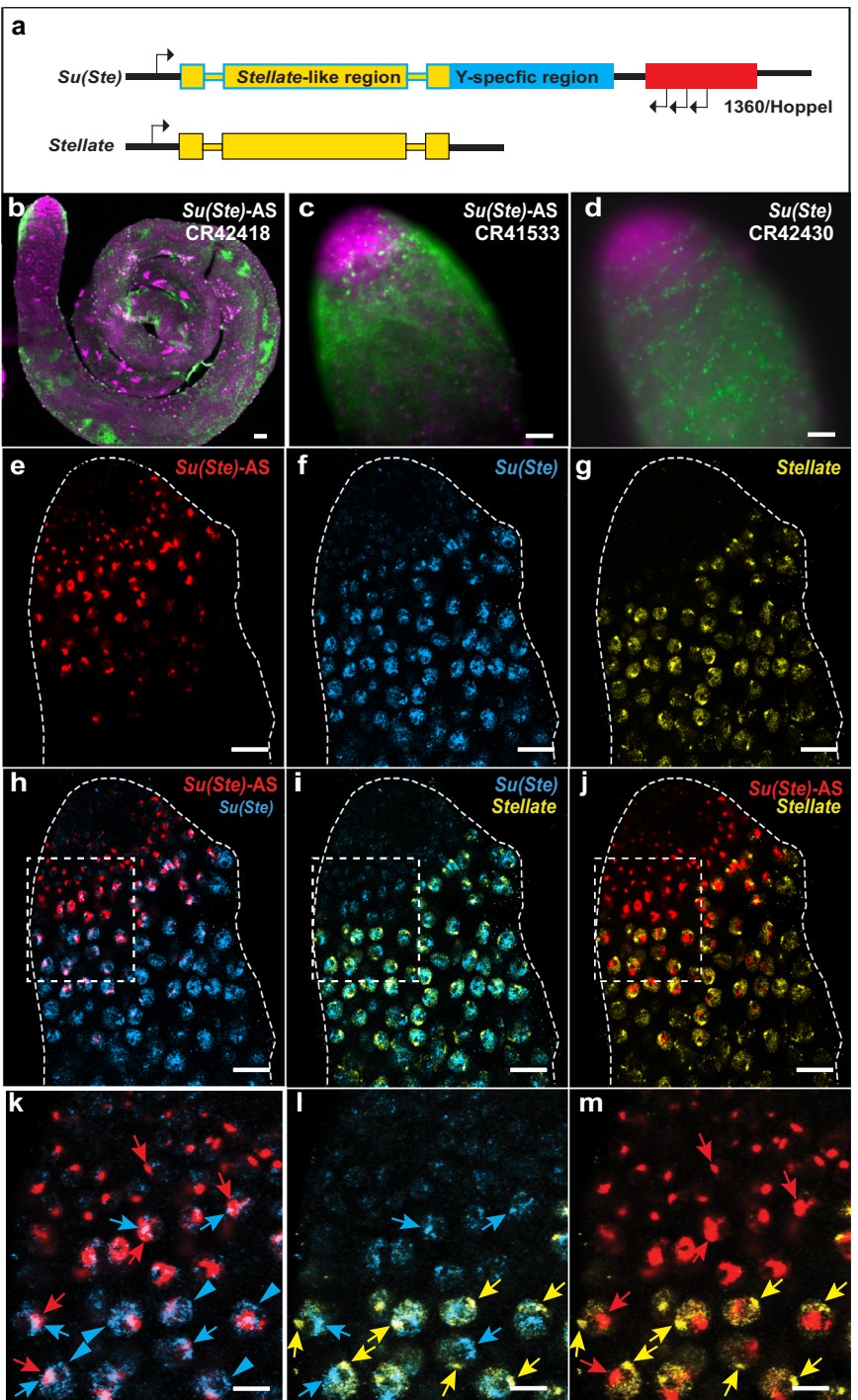

**Fig. 4 | *Su(Ste)* gene expression dynamics. a** Diagram of individual *Stellate* (*Ste*) and *Su(Ste)* repeat units. *Su(Ste)* repeats contain promoters at each end, allowing for bidirectional expression of complementary transcripts. *Su(Ste)* transcripts produced from the sense promoter are labeled as *Su(Ste)*, and transcripts produced by the *1360TE/Hoppel* TE are labeled as *Su(Ste)*-AS. **b** Widespread *Su(Ste)* antisense (*Su(Ste)*-AS, *CR42418*) transcript expression in testis. **c** Higher magnification of *Su(Ste)*-AS (*CR41533*) expression in gonialblast and spermatogonia nuclear foci and early spermatocyte cytoplasm. **d** *Su(Ste)* (*CR42430*) sense expression in spermatogonia nuclei and spermatocyte cytoplasm. **e**–**g** Expression patterns at the apical tip of the testis detected by HCR FISH using oligonucleotide probes that specifically target **e** *Su(Ste)-AS* (red), **f** *Su(Ste)* (blue) or **g** *Ste* (yellow). **h**–**j** Pairwise combinations of *Su(Ste)-AS, Su(Ste)* and *Ste*. **k**–**m** Higher magnifications of regions boxed in (**h**–**j**). Arrows indicate main nascent sites of expression. Blue arrowheads indicate minor sites of *Su(Ste)* localization that overlap with major sites of *Ste* localization. Scale bars = 20 µm.

functions. Both sense and antisense *Su(Ste)* transcripts are also detected in the seminal vesicle and ejaculatory duct lumens in interesting granule-like structures (Fly-FISH; Supplementary Fig. 4).

Due to the high similarity between *Su(Ste)* and *Ste* sense transcripts (~90% identity), we decided to also test shorter gene-specific

oligonucleotides probes using HCR FISH to ensure non-overlapping detection. As seen with our longer probes, the smaller probes also detect both sense and antisense *Su(Ste)* transcripts (Fig. 4e, f). Cytoplasmic transcripts were not detected, presumably due to the inability of the short probes to hybridize with processed piRNAs. Consistent

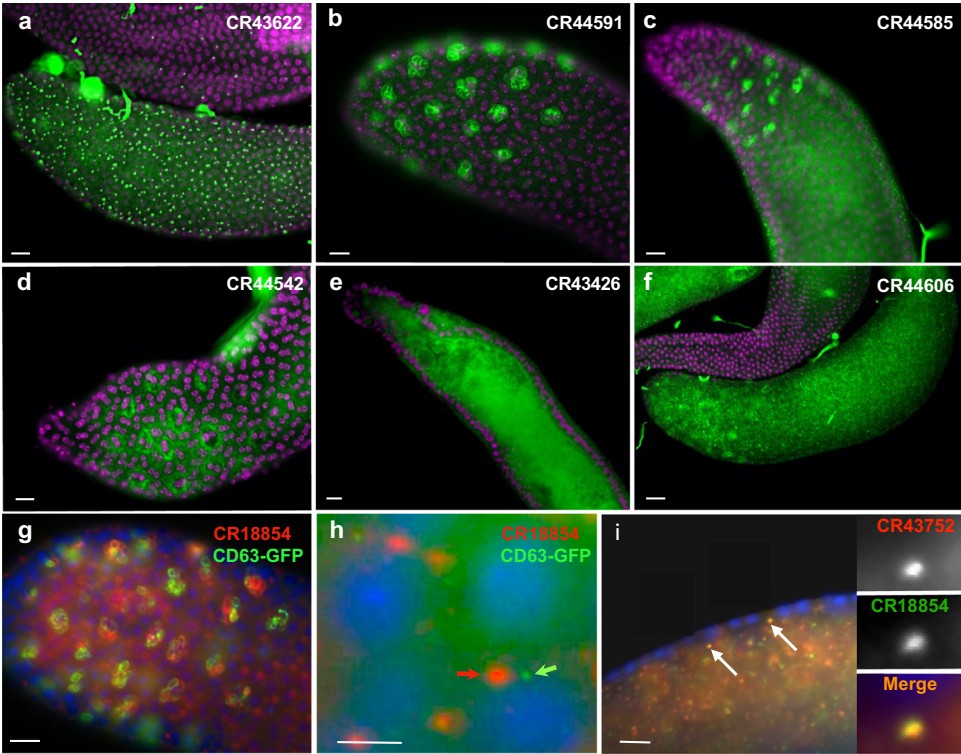

**Fig. 5 | lncRNA expression in accessory glands. a** *CR43622* RNA localized to single foci in main cell nuclei. **b** *CR44591* localized in multivesicular bodies in secondary cells. **c** *CR44585* localization in main cell cytoplasmic foci and secondary cell multivesicular bodies. **d** *CR44542* cytoplasmic localization in secondary cells. **e** Luminal localization of *CR43426* below the secretory cell layer. **f** *CR44606* localization in the cytoplasm of both main and secondary cells (and lumen – not shown). **g** Double-labeling of *CR18854* and CD63-GFP in the multivesicular bodies of secondary cells. Patterns appear to be more complementary than overlapping. **h** Higher magnification of *CR18854* and CD63-GFP-labeled particles in the accessory gland lumen. Particles are non-overlapping and differ in size. **i** Co-localization of *CR18854* and *CR43752* in accessory gland lumenal particles. Insets show higher magnification views. Scale bars = 20 μm.

with this assumption, HCR FISH detection of *Ste* transcripts in Y chromosome-deficient XO males yields abundant cytoplasmic signals (Supplementary Fig. 8). As expected, *Su(Ste)* signals are not detected in these XO testes.

As seen with full-length probes, HCR detection of *Su(Ste)*-AS transcripts shows earlier expression than the sense transcript. The antisense transcripts accumulate as single, large nuclear aggregates that most likely reflect transcripts accumulating at the *Su(Ste)* repeat locus (Fig. 4e, k) as previously noted[36]. The sense transcript, however, is dispersed throughout the nucleus in many smaller foci with the largest of these overlapping with the antisense transcript (Fig. 4f, l). HCR-FISH against *Ste* transcripts, shows a similar pattern of nuclear foci much like those of *Su(Ste)* sense transcripts, though mostly non-overlapping (Fig. 4g, m). Their appearance is delayed compared to both *Su(Ste)* transcripts. Notably, one of the smaller nuclear foci produced by *Su(Ste)* sense transcripts appears to co-localize with the major *Ste* transcript foci (Fig. 4k–m), suggesting possible function at the sites of *Ste* transcription. Taken together, our results suggest a major role for *Su(Ste)* sense transcripts in the generation of antisense piRNAs, as well as potential new roles in the nucleus.

### lncRNA expression in accessory glands

Accessory glands are relatively simple organs composed primarily of two types of secretory cells, main cells, and secondary cells, surrounding a hollow lumen that fills with secretory cell-produced seminal fluid. The majority of subcellular patterns observed here were in main or secondary cell cytoplasm and membrane regions, with most also found in the lumen (Fig. 5). One pattern that is predominant is within secondary cell multivesicular bodies (MVBs; Fig. 5b–d) as shown by double-staining for *CR18854* and the MVB marker CD63-GFP[37]

(Fig. 5g). Although both probes mark the same structures, their distributions are more complementary than overlapping.

Secondary cell MVBs have been shown to secrete vesicles referred to as exosomes, which can be visualized using a CD63-GFP fusion protein[37]. Since ~70% of the lncRNAs examined are in luminal puncta, we checked to see if these correlate with exosomes by double-labeling with exosome-marking[37] anti-GFP antibodies. As shown in Fig. 5h, the luminal puncta marked with CD63-GFP do not overlap with *CR18854*-containing puncta, which are also on average ~3X larger in diameter than GFP-labeled exosomes (200–500 nM vs.100–200 nM). We also looked to see if luminal lncRNA puncta tend to share similar lncRNAs or contain distinct lncRNA cargoes. As exemplified in Fig. 5i, all tested lumenal lncRNAs are within the same non-exosome particles (4/4 lncRNAs tested), although their relative amounts show some variability. Thus, it appears that these lncRNA puncta represent a new type of extracellular, lncRNA-enriched, particle.

To determine if this frequency of expression and secretion is also true for coding gene transcripts, we tested a randomly selected set of 96 coding genes (plate RA.303/AM40D from the DGC 2.0 cDNA collection). Only 7 of these are expressed in accessory glands, only 2 in MVBs and none in the lumen (Fly-FISH). Thus, RNA secretion into luminal particles appears to be lncRNA-specific.

## Discussion

Our analysis of lncRNA expression patterns via FISH confirms previous observations of relatively high numbers of genes expressed in testes and accessory glands, albeit at much higher levels than previously reported. The wide range of intricate cellular and subcellular patterns observed provide numerous insights into the potential origins and roles of this class of genes, the importance of which has been a matter of debate.

Our findings of both high frequency and high levels of lncRNA expression are a key takeaway, as previously reported low levels of expression have been pivotal in the argument that lncRNAs are mostly non-functional. Our analyses of lncRNA poly A signal sequences and polyadenylation show that prior low detection rates may be attributed, in part, to low levels of lncRNA polyadenylation, which appears to have no consequence on their accumulation or function. Another contributing factor to low detection levels may be that many lncRNAs are key components of phase-separated RNP structures, which have been shown to be inefficiently solubilized by conventional extraction methods[38]. Some other possibilities for low levels of lncRNA detection via RNAseq are incomplete reverse transcription due to lncRNA cleavage, processing or nicking, and the use of repeat masker software, which may eliminate the many lncRNAs with repetitive element-derived sequences[39] from further consideration.

As we found in a previous study of lncRNA expression[16], subcellular localization frequencies in male reproductive tissues are similar to those of mRNAs. Thus, subcellular trafficking is also the norm for lncRNAs, with correlative implications for co-localized lncRNA functions and mechanisms of action. Related to this point, the majority of well-studied lncRNAs have functions in the nucleus, leading many to assume that most functional lncRNAs act in the nucleus[3,17]. For example, the authors of a previous genetic analysis of lncRNAs in testes proposed nuclear functions for all of those required for fertility[23]. However, our analysis of these 32 required lncRNAs reveal nuclear patterns for only six, with five of those six also expressed cytoplasmically. As previously for other tissues[16], we found here that the majority of lncRNAs reside outside of the nucleus, most in the form of variable sized granules, reminiscent of previously described phase-separated structures.

Another important aspect of the lncRNA expression numbers and levels observed is the number that are expressed post-meiotically in differentiating spermatids. A recent single cell/single nucleus RNAseq study identified only ~160 coding genes (<0.5%) expressed post-meiotically[15]. In contrast, ~80% of lncRNAs analyzed here are found in spermatids, with the majority likely to be transcribed uniquely or most highly during these stages. This suggests that the expression of many lncRNAs may be driven by unique classes of promoters, transcription factors or polymerases. In terms of their potential function, they may be playing novel and important roles in sperm assembly, structure and function, or TE management. If true, this may also help explain why new lncRNA gene production and potential functions are tolerated, and perhaps facilitated (see below), during spermatogenesis.

In terms of whole genome sequencing and functional analyses, Y chromosomes represent one of the last frontiers due to their high content of TEs, satellite repeats and pseudogenes. While these repetitive elements render them predominantly heterochromatic and silent during other stages of development, their expression during spermatogenesis presents numerous challenges and dangers to developing sperm and progeny. Our results suggest multiple roles for lncRNAs in surmounting these challenges.

Among the 14 annotated Y chromosome coding genes, 6 contain extremely large introns composed primarily of extensive satellite repeats and TEs (*kl-5, kl-3, kl-2, Ppr-y, WDY*, and *ORY*). The large costs in terms of time, components and energy required to replicate, transcribe and process these giant 'mega-genes' suggest evolutionary advantages. Our observation of 28 mega-gene co-localizing lncRNA transcripts suggests potential roles for these lncRNAs in successful mega-gene production and processing. The fact that mutations in five of the eight Y-loop lncRNA genes previously tested for function affect fertility[23], with similar defects seen in late sperm bundle morphology and resolution[23,40], suggests that many of the other co-localizing lncRNAs will have similar functional requirements. These might include roles as sequence-specific guides that recruit additional RNAs or proteins needed for mega-gene transcript initiation, elongation

and/or splicing, or as nucleators or cross-linkers of phase-separating biomolecular condensates. The latter could help to compartmentalize, stabilize or functionally organize the enormous and potentially fragile Y-loop transcripts. Consistent with these suggestions, previous work has shown that antibodies that recognize triple helix nucleic acid structures, as well as antibodies directed against several RNA binding proteins, specifically mark Y-loop structures[25,26,28,41]. Here, we found that four Y-loop localizing lncRNAs with sequence-complementarity to one another, as well as to the Y chromosome mega-gene *Ppr-Y*, all co-localize to the same subnuclear domains. We also showed that another lncRNA not predicted to interact with *Ppr-Y*, but rather with *kl-3*, co-localizes with *kl-3* and not *Ppr-Y*.

Given that *Drosophila melanogaster* Y chromosome mega-genes evolved from autosomal genes with small introns[25,42], one possible explanation for their expansion is to promote heterochromatin formation in somatic cells where they are not needed and are potentially detrimental. As has been previously suggested[25], the resulting delay in transcript completion and processing may also ensure that protein accrual only occurs post-meiotically. The pre- or post-splice introns may also have additional functions.

In general, the presence of long satellite repeats can be detrimental to gene function and cell viability, due in part to their propensity for further expansion and to form insoluble condensates[43,44]. Thus, further functional analyses of lncRNAs that co-localize with these structures is likely to shed light on the normal handling of these long repetitive RNA sequences. Similar analyses may also help explain how large disease-prone genes in humans (e.g., *Dystrophin (DMD)*[45];) are efficiently transcribed and processed. These mechanisms will also pertain to human Y chromosome genes, many of which also contain large, repetitive element-rich, introns[46] and include newly discovered mega-lncRNA genes[47].

Another potential role for lncRNAs is in the arm's race that occurs between genes on the X and Y chromosomes. Our analysis of Y chromosome *Su(Ste)* gene repeats suggests new mechanisms for the management of X chromosome *Ste* genes. Initial studies suggested that *Ste* gene repression is mediated primarily by a ping-pong-based mechanism of piRNA production carried out by *Su(Ste)* antisense transcripts via pairing with *Ste* transcripts[32,36]. More recently, it has been shown that *Ste* targeting is initiated by maternally provided *1360/hoppel* TE fragments that recognize the 5' ends of *Su(Ste)* antisense transcripts[48]. Our demonstration of robust *Su(Ste)* sense gene expression prior to *Ste* gene expression suggests a major contribution to robust *Ste* antisense piRNA production prior to *Ste* gene expression. Our observation of unique nuclear localization patterns for both sense and antisense *Su(Ste)* transcripts also raises the possibility of parallel nuclear mechanisms of action. These inter-chromosomal interactions may be related to mechanisms that modulate similar processes such as meiotic drive, hybrid dysgenesis, speciation and progeny sex ratios.

The evolution of the *melanogaster Su(Ste)* system also exemplifies the type of TE-driven evolutionary processes that have been proposed to occur in testes[4–6]. Studies have shown that the initial Y chromosome *Su(Ste)* progenitor was subsequently modified by a *hoppel/1360* TE insertion at its 3' end, leading to the production of a new antisense transcript-producing lncRNA gene, which we refer to as *Su(Ste)-AS*. The amplification of these *Su(Ste)/hoppel* units led to the current system described here that controls *Ste* expression.

While TEs have many positive roles, such as driving new gene evolution, unrestricted expression would be disastrous. We show here that the lncRNA *flam*, previously shown to control TE expression maternally[30,31], is also expressed in the male germline. Probes directed against the *flam* first exon or first intron detect products with very different subcellular distributions, with the exon probe labeling cytoplasmic RNAs, and intron probes marking chromatin-associated complexes. This likely reflects different modes of action, with the spliced cytoplasmic products producing piRNAs and the nuclear

unspliced or intron products affecting TE gene chromatin state and transcript initiation.

This potential function of *flam* introns brings up a key point in lncRNA function—that many lncRNAs, like 'architectural' arcRNAs, may have roles and properties that are intron-dependent (reviewed in ref. 19). This is likely to be particularly true for lncRNAs that function in the nucleus, such as the Y-loop lncRNA *CR44206*, or others that have differing functions in the nucleus and cytoplasm.

In mammals, paternally expressed lncRNAs such as *H19, KCNQOT1*, and *Igf2* are required for the allelic inheritance of nearby coding gene expression states[49,50]. It has also been shown that sperm nuclei contain RNAs from both coding and noncoding genes, and that small RNAs derived from sperm nuclei can have trans-generational effects on metabolism and behavior[51,52]. Here, we observed several lncRNAs (e.g., *Rox1, flam*, and *Su(Ste)-AS*) on elongated spermatid nuclei. We also observed transcripts for many lncRNAs along developing and mature sperm tails, and many more were found in the lumens of seminal vesicles, ejaculatory ducts, and accessory glands (>70%). Taken together, these observations raise the possibility that many lncRNAs may provide paternal reproductive functions via a variety of mechanisms.

Our observation of ~40% of lncRNAs expressed in accessory gland secondary cell MVBs, with most also in lumenal puncta, suggested that they may contribute to exosome production or function. However, 4/4 lncRNAs tested do not co-localize with an exosome marker. Rather, all four are co-expressed within larger particles that likely represent a new type of secreted, lncRNA-rich vesicle or granule. Like exosomes[37], they may also play significant roles in the control of sperm activity, recipient female responses, and possibly embryonic development. If so, it is likely that these types of lncRNA particles will also be produced by other cell types, can modulate distant cell responses, and can provide insight into new variations of RNA-based therapies.

## Methods

### Probe production
Templates for probe production were made either from cDNA clones from the *Drosophila* Gene Collection (DGC) libraries generated by the Berkeley *Drosophila* genome project or from PCR amplicons. Amplicons were made by designing 20-mer primers complementary to lncRNA gene sequences. Sequences for T7 and T3 polymerase binding and universal primer use were added virtually to either end prior to ordering (T7-oriented for antisense probe production). Digoxygenin-labeled RNA probes were generated from templates as previously described[53].

In cases where HCR FISH was used, split oligonucleotide probes (listed in Supplemental Table 2) were ordered from Eurofins. Hairpin amplification primers with indicated ALEXA tags were purchased from Molecular Imaging.

### TSA-FISH protocol
$w^{1118}$ adult flies were reared at room temperature on standard cornmeal food. Unless stated otherwise, males used for dissection were 3 days old and unmated. Tissues were collected, fixed, hybridized, and developed for signals as described[54,55] using Tyramide signal amplification (TSA) to develop probe signals.

### HCR FISH
The hybridization chain reaction (HCR) FISH protocol used[56] was optimized for use with *Drosophila* testes with the following modifications. In most cases, testes were cut in half while in PBS, then fixed on polylysine-coated slides as previously described[57]. When not used immediately, sample-containing slides were stored at 4 °C in 95% ethanol and, prior to use, rinsed 3× for 5 min in PBS. For hybridization, dissected tissues were surrounded by rectangular parafilm cut-outs to contain hybridization solutions and pre-hybridized in "30% hybridization buffer"[56] for 30 min at 37 °C. Probe solutions were prepared by combining 67 μl of 1.5× hybridization buffer (45%), 4 μL of each 20 μM

probe-set and ultrapure water to a total volume of 100 μL. Prior to adding probe, the slide-mounted tissues were shifted to 80 °C for 5 min in hybridization buffer and then 5 min with pre-heated probe solution. The slides were then incubated at 37 °C overnight in closed boxes containing pre-wetted Kimwipes to prevent dehydration. After hybridization, slides were rinsed once with pre-warmed (37 °C) probe wash buffer, then washed three times for 20 min at 37 °C with wash buffer, and finally 30 min at room temperature in amplification buffer. Prior to amplification, 3 μM fluorescently labeled hairpin amplification stock solutions were heated for 90 s at 95 °C and cooled at room temperature for 30 min in a dark drawer. Working hairpin solutions were prepared by adding 50 μl of 2× amplification buffer, 3 μL each of individual cooled hairpin pairs, and ultrapure water to a total volume of 100 μL. Samples were incubated with hairpin amplification mixtures at room temperature overnight in closed boxes in the dark. On the third day, samples were rinsed and washed four times for 5 min with 5× SSCT (SSC plus 0.1% Tween-20), three times for 5 min with PBS, and then mounted using DAPI-containing mounting media (EMS, Cat#17985-51).

### TSA-FISH double-labeling
The majority of RNA-RNA and RNA-protein double labeling was performed as described in[58], with the following modifications. 0.1% Tween-20 and 0.03% Triton X-100 were used for RNA-protein double-labeling. The following antibodies were used: rabbit anti-GFP (1:400, Abcam # 290), mouse anti-S5 (1:2, obtained from Harald Saumweber), Alexa488-conjugated goat anti-mouse (1:000; Invitrogen #A-11001) and FITC-conjugated donkey anti-rabbit (1:1000; Jackson ImmunoResearch #711-095-152). The CD63-GFP expressing fly stock, A17122 (w; *spitz-GAL4, tub-GAL80ᵗˢ; UAS CD63-GFP/SMS TM6*), was provided by Clive Wilson.

### Microscopy and imaging
All FISH images were captured using a Leica DMRA2 epifluorescence microscope, or occasionally a Leica SP8 Lightning Confocal/Light Sheet or Leica SP8 Lightning Confocal/STED microscope (indicated in Fig. legends). All images were processed using Photoshop and ImageJ and are publicly available on our Fly-FISH database (www.fly-fish.ccbr. utoronto.ca).

### lncRNA complementary site searches
Complementary sites between Y-loop lncRNAs were identified by mapping lncRNA fragments of length k (19 to 25 bp), referred to as *k*-mers, to full-length Y-loop lncRNA sequences. Sequences for the 28 Y-loop lncRNAs were retrieved from Flybase (flybase.org). *K*-mers of each lncRNA sequence were generated using a sliding window algorithm to produce all possible *k*-mer combinations, where the total number of *k*-mers per lncRNA equals lncRNA length minus *k*-mer length plus one. lncRNA *k*-mers were grouped into datasets by *k*-mer length. Using the small read mapper Bowtie 1.3.1 (https://sourceforge. net/projects/bowtie-bio/files/bowtie/1.3.1/), each *k*-mer dataset was queried against the pool of 28 Y-loop lncRNA sequences. Each dataset was queried three times with a varying number of allowable mismatches ranging from zero to two mismatches. Bowtie 1 match files were converted to BED file format and filtered for antisense matches.

### Polyadenylation signal sequence identification
All annotated transcript sequences for mRNA and lncRNA PAS searches were retrieved from Flybase under the bulk file names "dmel-all-transcript-r6.51 fasta" and "dmel-all-ncRNA-r6.51.fasta", respectively. Both groups of sequences were then filtered to include transcripts greater than 200 nucleotides and containing no unannotated nucleotides in the search windows. PAS variants reported in ref. 59 were searched within the last 50 nucleotides of each sequence using Python. In cases where multiple PAS variants were found in one transcript, the

most commonly used variant was assigned to that gene. To search for DSEs, 50 nucleotide sequences downstream of each transcript 3' terminus were derived from the genome file, "dmel-all-chromosome-r6.51.fasta" and searched using regex for combinations of nucleotides found 10% or more of the time in the DSE sequence logo reported in ref. 60.

## RT-PCR

Testes of 0-4d old virgin males were dissected in PBS within a 15 min time interval. Total RNA was extracted using a GeneJET RNA Purification Kit (Thermo Scientific, Cat No: K0731) according to the manufacturer's instructions, with DNase I (Thermo Scientific, Cat No: EN0521) added to 0.1 U/μl. Tissue lysate was homogenized by passing through a 20-gauge syringe needle 100X[61] and stored at −70 °C if not used immediately. Reverse transcription was performed at 52 °C using SuperScript III (Invitrogen, Cat No: 18080093) as described by the manufacturer. In all, 1 μg total RNA was reverse transcribed using either Oligo (dT)$_{12-18}$ (Invitrogen, Cat No.: 18418012) or gene-specific primers (listed in Supplemental Table 1) to initiate transcription. PCR was performed using an Applied biosystems SimpiAmp thermal cycler with the following program: 95 °C for 3 min, then 35 cycles of 95 °C, 45 s, 55 °C, 45 s, 72 °C, 45 s, and a final extension at 72 °C for 5 min.

## Statistics and reproducibility

All patterns shown here and uploaded on Fly-FISH are representative of typical patterns seen in multiple specimens, multiple experiment repeats (2–5), and in some cases, with different probes or FISH methods. Variations of patterns can be found on FlyFISH.

## Reporting summary

Further information on research design is available in the Nature Portfolio Reporting Summary linked to this article.

## Data availability

The data supporting the findings of this study are available from the corresponding authors upon request. All in situ images acquired, as well as corresponding annotations, can be accessed on the Fly-FISH database (https://fly-fish.ccbr.utoronto.ca/). Search tools facilitate selection by gene name, localization term and combinations thereof. Source data are provided with this paper.

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

## Acknowledgements

The authors would like to thank Harald Saumweber for kindly providing S5 antibody and Clive Wilson for providing the UAS-CD63-GFP reporter fly line. We are grateful to Paul Paroutis and Kimberly Lau at the SickKids Imaging Facility for assistance with confocal microscopy. This work was funded by the Canadian Institutes of Health Research Project grants provided to H.M.K. (PJT-165885), to H.M.K. and J.A.B. (PJT-165884), and by NSERC Discovery grants awarded to J.A.B. (RGPIN-2016-06775 and RGPIN-2022-05163).

## Author contributions

Conceptualization: H.M.K.; performed research: Z.S., J.H., C.H., R.W., A.J., M.J., and L.M.; data interpretation and analysis: Z.S., R.W., A.J., M.J., A.S., I.D., P.S.-L., J.A.B., and H.M.K.; figure and table preparation: Z.S., M.J., and H.M.K.; writing, reviewing, and manuscript editing: H.M.K., J.A.B., and M.J.; funding: J.A.B. and H.M.K.

## Competing interests

The authors declare no competing interests.
