## [Peer Review File · Nature Communications]

Spatially revealed roles for lncRNAs in *Drosophila* spermatogenesis, Y chromosome function and evolutionREVIEWER COMMENTS

Reviewer #1 (Remarks to the Author):

In the manuscript title "Extensive roles for lncRNAs in male reproductive tissues", Shao and colleagues seek to uncover the tissue-specificity and subcellular-localization of *Drosophila* lncRNAs. This study aims to increase localization and expression specificity of several lncRNAs to add new resource to both the *Drosophila* and lncRNA research communities. Interestingly, the authors provide evidence for nuclear and exported lncRNAs that are highly expressed in distinct spatial patterns, a novel type of secreted lncRNA structure, the potential role of lncRNA in regulating the Y-chromosome mega-gene transcripts and X- and Y-chromosome transposable elements (TEs), as well as paternal inheritance and evolution. While the work is interesting, the manuscript's relies on RNA-FISH methods and lacks functional validation that limit the conclusions and overall impact of the work. Furthermore, although the manuscript is well-written and easy to follow, the data presented are highly disorganized and would require substantial revision for clarity.

Major comments

1- An important takeaway of the manuscript is that RNA-FISH shows strong expression patterns for many lncRNAs that are likely to be essential. However, these data rely on qualitative RNA-FISH expression and no information is provided on their essentiality and function. Also, information on how the assays were standardized across different lncRNAs will be important. It may be good to validate, standardize, and quantify the results using alternative approaches.

2- The authors argue that Poly A sequences may explain differences in RNA-seq vs. RNA-FISH coverage of lncRNAs (Figures 1-D and E, Extended Data Figure 1-A, and text on page 4, lines 19-20) - The differences presented in these results seem to be explained by the typical "PAS+" sequence, AATAAA, occurring less frequently in lncRNAs than mRNAs. The authors may wish to be more cautious to conclude typical poly A sequences do not seem necessary for lncRNA expression and function. Additionally, molecular methods such as amplification-free sequencing may help uncover potential polyadenylation differences in lncRNAs vs. mRNA and "PAS+" vs. "PAS-" lncRNAs.

3- Correlations between lncRNA expression and function (Figures 2-5): While these studies provide evidence for associations between lncRNA expression and function, these studies could be substantially improved by adding complementary molecular functional validation methods including lncRNA genetic knockdowns or RNAi to validate the regulatory roles of the identified lncRNAs. Similarly, a suite of immunoprecipitation tools coupled with mass spectroscopy (MS) or next-generation sequencing, such as Chromatin Isolation by RNA Purification (ChIRP)-MS or -seq, could be used to examine the extensive potential for interactions of candidate lncRNAs with DNAs, other RNAs, and/or proteins to help unravel their functional roles at the mechanistic level.

4- As mentioned above, the manuscript seems disorganized. A few examples are presented below:

a) Figure 4 legend and text (page 10, lines 3-13): The figure legend for Figure 4A-D seems to be mixed up, and the figures referenced in the text seem out of order. Additionally, data in Figure 4D do not seem to clearly show early nuclear and late cytoplasmic accumulation of Su(Ste) sense lncRNA transcripts.

b) Extended Figures: Many of the extended figures are cited incorrectly and out of order in the text.

Overall, while the manuscript is interesting, the results are preliminary and would benefit additional validation for at least 1-2 novel lncRNAs identified in this study.

Reviewer #2 (Remarks to the Author):

The manuscript entitled "Extensive roles for lncRNAs in male reproductive tissues" represents an extensive characterization of lncRNA expression in *Drosophila* male germline. This study also involves many comparisons to protein coding genes further providing important information in how they compare and contrast. For example, they demonstrate, that poly adenylation signals for lncRNA are vastly differ to mRNAs -- the first report of this to this reviewers knowledge. Moreover, an important insight for future lncRNA studies as many important lncRNAs may have gone missing due to lack of polyA targeting, common to most sequencing approaches (similar to the well established and function NORAD lncRNA in mammalian cells).

To overcome this limitation the authors use RNA-FISH that as a systematic approach to understand lncRNA expression in the male reproductive tract. This approach led to more detailed insights into lncRNA abundance estimates that were much higher than observed by RNAseq. Moreover, the common finding that lncRNAs are more nuclear than mRNAs was also overturned in this study by using RNA-FISH. Finally, and most impressively the authors find a correlation between abundance and sub cellular localization that indicated functionality in genetic loss of function studies - something that will greatly facilitate the identification of functional lncRNAs from transcriptional noise.

The authors continue to examine lncRNAs within Y-loops (genes with very large introns - or mega genes) and overlap with several key RNA processing genes. Interestingly, the authors find that lncRNAs on other chromosomes show sequence similarity to those within Y-loops. This is suggestive of cross-chromosomal RNA-RNA interactions to the Y-chromosome. Indeed, the authors demonstrate that CR44206 has complementary sequences to three Y-loop transcripts (CR43193, CR44619 and CR45805). Next the authors examine the interaction of noncoding RNA and transposable element silencing. Specifically, the Flaminco (*flam*) gene that is a transcriptional precursor to piRNAs that in turn silence transposable elements - typically thought to cytoplasmic. However, the RNA-FISH experiments revealed that some of the *flam* processed RNA sequences are nuclear. Even more tantalizing these lncRNAs are expressed in seminal vesicle lumens, perhaps suggestive of paternal RNA inheritance. In another example the authors examine the *Su(Ste)* transcripts that includes an antisense RNA that was previously thought to be lowly expressed before examination in this study by RNA-FISH. Interestingly the processed products of the *Su(Ste)* antisense transcript were also found in the elongating spermatid nuclei. Thus, similar to *flam* maybe paternally inherited. Notably, the authors were concerned about possible sequence similarity issues confounding the RNA-FISH approach and performed appropriate controls to ensure the expression patterns observed. This further leads to a hypothesis that the protein coding mRNA may be a primer for generating non-coding RNAs based on the temporal dynamics of expression between the sense and anti-sense transcripts.

Overall, there are many interesting and important findings for lncRNA biology in this study - including the importance of the RNA-FISH approach in this study. I don't some comments below for then authors consideration.

1) The results read a bit like a discussion including a lot of background information. I do find this important for the general readership of Nature Communications to put these results in the context of previous results in *Drosophila*. However, I wonder if the authors could streamline the results somewhat and then include some of the discussion in the results section into the discussion.

2) There is some concern about cross-hybridization of the probes used in this study as indicated on page 9 line 23.

It maybe important to try and use an orthogonal RNA-FISH approach (e.g., MERFISH). Many of these important results depend on non-cross hybridization and thus worth trying smaller probes for a percentage of the lncRNAs examined (as was done for *Su(Ste)*). It is suggested that the authors take a random selection of 1-10% of the transcripts examined and use an orthogonal RNA-FISH method with smaller probes that may be less susceptible to cross-hybridization.

3) It is also advised to perhaps perform a non RNA-FISH method to validate the global RNA-FISH patterns presented. For example, RT-qPCR using random hexers (to avoid the noted poly-A issues in expression measurements) to create cDNA and measure abundance in the male reproductive tissues. This perhaps could be done by use fluorescent markers to isolate the same samples presented and measure abundance. This again could be done with 1-10% of random selected examined lncRNAs to compare to the RNA-FISH results.

While it is expected orthogonal methods may not validate all the results by RNA-FISH it may provide the reader with more assurance that all results are not based on one method. Moreover, a "confidence" level for re-examination of specific lncRNAs presented in this study using orthogonal methodologies. Yet this reviewer understands that there are many lncRNA studies based on one approach (e.g., RNAseq). Thus, it is suggested but perhaps outside the scope to validate the expression patterns reported by at least one or two orthogonal methods -- especially considering the importance of the implications of the findings in this study.

4) It would be very interesting if the authors could examine if noncoding RNA is paternally inherited. This would be of great interest to the readership of Nature Communications. Perhaps this implication could be further evidenced by a female KO of one of these genes and perform RNA-FISH on the maternal abundance of these RNAs upon fertilization.

5) Perhaps the title should focus on expression patterns rather than "roles" which implies function. While it is understandable that the authors correlate their expression findings with genetic studies that truly imply function - yet the title has an initial implication that this is a comprehensive genetic functional screen. Moreover, it maybe worth mentioning this study is in Drosophila as it maybe distinctive from mammalian male reproductive tissues - as many lncRNAs are not conserved as deep as Drosophila. This is not to discount the key results in this study but to distinguish evolutionary distances of equal importance.

Reviewer #3 (Remarks to the Author):

Shao's paper determined the subcellular localization of a number of lncRNAs in fly male reproductive organs mainly by using in situ hybridization. The imaging quality is good and the work can provide some clues to further functional and mechanism studies of individual lncRNAs. So, it looks fine as a resource paper. However, the research is focused only on a restricted area and lacks deeper studies, thus, it hardly arouses broad interest. I do not think there are enough innovative findings to be published in Nature Communications. Please see my specific comments below.

1. In introduction part, the description of both lncRNA and spermatogenesis is superficial. I could not get much information there. Important lncRNA function and action mechanism, and some critical pathways in spermatogenesis should be described.

2. Authors proposed many possibilities but could not prove them.

Response to reviews:

We thank the reviewers for their helpful comments and suggestions, which have allowed us to strengthen our study and manuscript significantly. While the second reviewer was quite enthusiastic about our numerous findings, the first and third reviewers had concerns about the depth of our follow-up analyses. We think this may have been due in part to not fully appreciating our incorporation of the many functional characterizations already done by others, including CRISPR and RNAi studies, that nicely validate the relevance and nature our findings, hypotheses and conclusions. The details of these complementary studies are provided in our responses to each of the reviewers below. As a reminder, among our newly reported general findings that will have a huge impact are 1) that ~85% of lncRNAs are non-nuclear, with a vast array of cytoplasmic and extracellular distributions, 2) that the majority of lncRNAs appear not to be polyadenylated, 3) that levels of expression are much higher than forecast by RNAseq, 4) that the majority of lncRNAs continues to be transcribed after spermatocyte meiosis, when less than 0.5% of coding genes are transcribed, and 5) that various analyses suggest that the majority of lncRNAs are functional. These findings are all contrary to prior assumptions. Our further demonstrations of likely major roles for lncRNAs in Y chromosome mega-gene processing, transposable element management, meiotic drive and paternal contributions to embryogenesis are also highly novel and will have large impacts. As rightly pointed out, this is also a resource paper with thousands of new data points that will help countless other researchers. The associated database, FlyFISH, is already linked to and from FlyBase, the major *Drosophila* database and, without the testes data yet published, already receives an average of ~500 hits per day.

Reviewer 1

In the manuscript title “Extensive roles for lncRNAs in male reproductive tissues”, Shao and colleagues seek to uncover the tissue-specificity and subcellular-localization of *Drosophila* lncRNAs. This study aims to increase localization and expression specificity of several lncRNAs to add new resource to both the *Drosophila* and lncRNA research communities. Interestingly, the authors provide evidence for nuclear and exported lncRNAs that are highly expressed in distinct spatial patterns, a novel type of secreted lncRNA structure, the potential role of lncRNA in regulating the Y-chromosome mega-gene transcripts and X- and Y-chromosome transposable elements (TEs), as well as paternal inheritance and evolution. While the work is interesting, the manuscript’s relies on RNA-FISH methods and lacks functional validation that limit the conclusions and overall impact of the work. Furthermore, although the manuscript is well-written and easy to follow, the data presented are highly disorganized and would require substantial revision for clarity.

First, we are happy that the reviewer found the manuscript well written. Although the reviewer also stated that the data presented was highly disorganized, they pointed out only a few instances of this. These are commented on below in our response to the reviewer’s specific examples.

Regarding the “lack of functional validation”, please see our initial response above and the more detailed responses to point 3 below.

Major comments

1- An important takeaway of the manuscript is that RNA-FISH shows strong expression patterns for many lncRNAs that are likely to be essential. However, these data rely on qualitative RNA-FISH expression and no information is provided on their essentiality and function. Also, information on how the assays were standardized across different lncRNAs will be important. It may be good to validate, standardize, and quantify the results using alternative approaches.

We will respond to the issues of function further below (point 3, for example). In terms of assay rigor and reproducibility, we note the following. First, approximately one quarter of the ~600 genes studied here were tested multiple times with multiple probes and in multiple experiments with only rare discrepancies in terms of spatial or quantitative results. Second, each set of genes assessed (up to 90 at a time) includes a pair of positive and negative internal controls. Third, in numerous experiments we have also used the oligonucleotide-based HCR FISH and smFISH approaches with highly similar results. In fact, the latter two tend to suffer from issues that RNA probe-based FISH does not, such as the off-target binding of a subset of oligos, and failure of some oligos to bind sequences that are masked by proteins or intra/inter RNA sequence interactions. As shown in Fig. 4 and extended Fig. 8, oligonucleotide probes are also less likely to detect processed RNAs such as piRNAs.

In addition to our dozens of FISH-based studies prior to this one, we have also published ten methods papers, mostly invited, making us reasonable experts in the field. Additional support for this statement is that our FISH expression database FlyFISH, which documents the developmental expression of approximately 10,000 genes at the subcellular level, is linked in and out of FlyBase, the go-to portal for *Drosophila* gene information. In the recent fly community consortium analyses of whole fly and tissue-specific snSEQ data, we were brought on board as the main spatial expression experts to independently validate the spatial and quantitative aspects implied by the sequence-based testis expression data (results published recently in *Science* and *eLife*). While validating many of the findings, we also exposed a number of incorrect assumptions.

Although the RNA probe-based FISH methodology that we employed is not considered as quantitative as single molecule FISH (smFISH), it comes very close, and if carried out properly, is quite accurate. The major things that can affect signal intensity using our approach are probe length and concentrations. To control for this, we use an average probe length of ~800 nucleotides, and transcribed probes are always examined on agarose gels for correct length and abundance, with amounts used adjusted accordingly. Nevertheless, we opted not to push our evaluations of expression levels too far. Comparisons were made to the well characterized internal controls and limited to the terms “strong, moderate, weak or not detected”.

2- The authors argue that Poly A sequences may explain differences in RNA-seq vs. RNA-FISH coverage of lncRNAs (Figures 1-D and E, Extended Data Figure 1-A, and text on page 4, lines 19-20) - The differences presented in these results seem to be explained by the typical "PAS+" sequence, AATAAA, occurring less frequently in lncRNAs than mRNAs. The authors may wish to be more cautious to conclude typical poly A sequences do not seem necessary for lncRNA expression and function. Additionally, molecular methods such as amplification-free sequencing may help uncover potential polyadenylation differences in lncRNAs vs. mRNA and "PAS+" vs. "PAS-" lncRNAs.

There were a couple of points made in this comment. For the first, we are very confident in the numbers of lncRNAs that do not contain poly A signals. These numbers are similar to what have been suggested previously for *Drosophila* and vertebrate lncRNA genes, although based on outdated and limited analyses. We've seen 100% correlation so far between RNAs predicted to contain or not contain poly A tails when reverse transcribing with oligo dT vs. 3'-specific primers, as demonstrated in Fig 1h. This turned out to be the case for all of the lncRNAs tested that showed high abundance via FISH and very low/non detection by RNAseq. For coding genes, our quantitation data correlate very well with expression levels reported via RNAseq and scSEQ data. It is only the lncRNAs that show this extensive discord.

Regarding the reviewer's comment that we should "be cautious to conclude typical poly A sequences do not seem necessary for lncRNA expression and function", if the reviewer meant that there may be functional cryptic PAS elements for some, or that many lncRNAs will/do require a poly A tail, we agree. Clearly, at least half of lncRNAs are polyadenylated. However, our data (and that of others) show quite conclusively that a significant portion of lncRNAs do not have poly A tails and yet manage to accumulate and function, indicating alternative mechanisms for achieving properties such as transcript stabilization (e.g.: some form triple helix structures or have protein binding motifs).

3- Correlations between lncRNA expression and function (Figures 2-5): While these studies provide evidence for associations between lncRNA expression and function, these studies could be substantially improved by adding complementary molecular functional validation methods including lncRNA genetic knockdowns or RNAi to validate the regulatory roles of the identified lncRNAs. Similarly, a suite of immunoprecipitation tools coupled with mass spectroscopy (MS) or next-generation sequencing, such as Chromatin Isolation by RNA Purification (ChIRP)-MS or -seq, could be used to examine the extensive potential for interactions of candidate lncRNAs with DNAs, other RNAs, and/or proteins to help unravel their functional roles at the mechanistic level.

As pointed out in the manuscript, RNAi knockdown and CRISPR KO have already been conducted by others for many of the lncRNAs examined in our study. Examples include 8 of our 30 'Y-loop' lncRNAs, along with 100 other lncRNA genes that were characterized by Wen et al, (2016). However, while these knockouts were shown to

affect spermatogenesis and fertility, the underlying mechanisms were not addressed. Our study provides deep insight into why those KOs had the effects they had (e.g.: interactively regulating Y chromosome mega-genes). Indeed, we now note that 5 of the 8 Y-loop lncRNA genes knocked out exhibited highly similar phenotypes to the kl-2,3,5 mega gene mutations, with effects on late elongation sperm nuclear bundle morphology and subsequent sperm tail individualization. Our observation that these 5 lncRNAs also have high sequence conservation among *Drosophila* species is consistent with these functions being highly conserved, whereas the other 3 of the 8 tested that did not show obvious phenotypes may have redundant or evolving functions. Other examples of lncRNAs that have already been tested genetically include the *Su(Ste)* genes, *flamenco* (but not in the male germline), *iab-4,8* (Hox complex lncRNAs), *HSR omega* (stress response), *Rox1,2* (dosage compensation; also published by us separately), *acal* and many others.

One of the likely reasons for the reviewer's concern is that our study is not the more typical type that focuses on one gene or process, and then drills down with multiple approaches. As noted by reviewer 3, our study is more of a resource type, perhaps more comparable to the recently published "Fly cell atlas" *Science* article (2022) describing the data obtained from extensive snSEQ data. Notably, we published a FISH-based study similar to this submitted manuscript a number of years ago on coding genes that has now been cited over 1,070 times (Lecuyer et al, *Cell* **131**, 2007). These types of papers have huge impacts. Both of the studies mentioned above had far fewer follow-up experiments and analyses than the current manuscript.

4- As mentioned above, the manuscript seems disorganized. A few examples are presented below:

- a) Figure 4 legend and text (page 10, lines 3-13): The figure legend for Figure 4A-D seems to be mixed up, and the figures referenced in the text seem out of order. Additionally, data in Figure 4D do not seem to clearly show early nuclear and late cytoplasmic accumulation of *Su(Ste)* sense lncRNA transcripts.
- b) Extended Figures: Many of the extended figures are cited incorrectly and out of order in the text.

Thank you for pointing out these figure labeling errors. For Fig. 4, this happened when the diagram in panel a) was moved from an earlier position as panel d) to its more logical introductory position. Moving the corresponding legend text from d) to a) has fixed the discrepancy.

The reviewer was correct that "Figure 4D" (now c) does not show later *Su(Ste)* gene expression. Since the later patterns are not relevant to the points discussed here, we now refer readers to our Fly-FISH database if they wish to explore these patterns further.

In terms of the 'many' additional extended figure incorrect citations, we have corrected all improperly cited figures (Ext data Fig 1 was out of place). We have also changed the term Extended data figure to Supplemental figure, as appears to be the norm for *Nat comm*.

Overall, while the manuscript is interesting, the results are preliminary and would benefit additional validation for at least 1-2 novel lncRNAs identified in this study.

Please see prior responses. Many of the genes assessed here have already been tested for functionality.

Reviewer 2

The manuscript entitled "Extensive roles for lncRNAs in male reproductive tissues" represents an extensive characterization of lncRNA expression in *Drosophila* male germline. This study also involves many comparisons to protein coding genes further providing important information in how they compare and contrast. For example, they demonstrate, that poly adenylation signals for lncRNA are vastly differ to mRNAs -- the first report of this to this reviewers knowledge. Moreover, an important insight for future lncRNA studies as many important lncRNAs may have gone missing due to lack of polyA targeting, common to most sequencing approaches (similar to the well established and function NORAD lncRNA in mammalian cells).

To overcome this limitation the authors use RNA-FISH that as a systematic approach to understand lncRNA expression in the male reproductive tract. This approach led to more detailed insights into lncRNA abundance estimates that were much higher than observed by RNAseq. Moreover, the common finding that lncRNAs are more nuclear than mRNAs was also overturned in this study by using RNA-FISH. Finally, and most impressively the authors find a correlation between abundance and sub cellular localization that indicated functionality in genetic loss of function studies - something that will greatly facilitate the identification of functional lncRNAs from transcriptional noise.

The authors continue to examine lncRNAs within Y-loops (genes with very large introns - or mega genes) and overlap with several key RNA processing genes. Interestingly, the authors find that lncRNAs on other chromosomes show sequence similarity to those within Y-loops. This is suggestive of cross-chromosomal RNA-RNA interactions to the Y-chromosome. Indeed, the authors demonstrate that CR44206 has complementary sequences to three Y-loop transcripts (CR43193, CR44619 and CR45805). Next the authors examine the interaction of noncoding RNA and transposable element silencing. Specifically, the Flaminco (flam) gene that is a transcriptional precursor to piRNAs that in turn silence transposable elements - typically thought to cytoplasmic. However, the RNA-FISH experiments revealed that some of the flam processed RNA sequences are nuclear. Even more tantalizing these lncRNAs are expressed in seminal vesicle lumens, perhaps suggestive of paternal RNA inheritance. In another example the authors examine the Su(Ste) transcripts that includes an antisense RNA that was previously thought to be lowly expressed before examination in this study by RNA-FISH. Interestingly the processed products of the Su(Ste) antisense transcript were also found in the elongating spermatid nuclei. Thus, similar to flam maybe paternally inherited. Notably, the authors were concerned about possible sequence similarity issues confounding the RNA-FISH approach and performed appropriate controls to ensure the expression patterns observed. This further leads to a hypothesis that the protein coding

mRNA may be a primer for generating non-coding RNAs based on the temporal dynamics of expression between the sense and anti-sense transcripts.

We thank the reviewer for the insightful takes on our results, though we note a minor inconsequential mix-up in the last few lines above (the *Su(Ste)* sense and antisense transcripts were mixed up, and both are noncoding). We also got confused in the early stages of this study.

Overall, there are many interesting and important findings for lncRNA biology in this study - including the importance of the RNA-FISH approach in this study. I don't have some comments below for the authors' consideration.

1) The results read a bit like a discussion including a lot of background information. I do find this important for the general readership of Nature Communications to put these results in the context of previous results in *Drosophila*. However, I wonder if the authors could streamline the results somewhat and then include some of the discussion in the results section into the discussion.

Great suggestion. We found a couple of places where the length of introductory information could be streamlined, as well as the conclusions. On the other hand, as noted by reviewer 3, a minimal amount is important for non-specialists to understand the issues. We hope the reviewer finds the new manuscript version to be more succinctly written.

2) There is some concern about cross-hybridization of the probes used in this study as indicated on page 9 line 23.

It may be important to try and use an orthogonal RNA-FISH approach (e.g., MERFISH). Many of these important results depend on non-cross hybridization and thus worth trying smaller probes for a percentage of the lncRNAs examined (as was done for *Su(Ste)*). It is suggested that the authors take a random selection of 1-10% of the transcripts examined and use an orthogonal RNA-FISH method with smaller probes that may be less susceptible to cross-hybridization.

We have already done what the reviewer requests, here and in previous studies. For example, as noted by the reviewer, we have used HCR FISH for *Su(Ste)* and *Ste*, as well as ~20 of the Y-loop lncRNAs, 6 Y-chromosome mega genes and ~40 coding genes. We have also tested hundreds of genes multiple times with similar or different RNA-based probes covering different genetic regions. In all cases, the results were the same, with the exception of a few HCR FISH oligo probes that gave no signals or, instead, gave high levels of broadly distributed background (presumably off-target). In total, our group has analyzed the expression of approximately 10,000 genes in our efforts to catalogue the cellular and subcellular distributions of all genes encoded in the *Drosophila* genome. In one of our published controls, we generated probes from *wingless* genes derived from 12 different *Drosophila* species that range in sequence similarity from 96% to 71%. Under our FISH conditions, the signals detected by probes derived from these orthologues dropped by i) ~50% for the 96% homologous orthologue, ii) 90% for the 91% orthologue and iii) 99% for the 84% orthologue. These

drops are despite all of these orthologues having short regions of near 100% homology. In short, there is more danger of short oligos yielding off-target signals than longer RNA probes. The specific mention by the reviewer of the potential problems with *Su(Ste)* and *Ste* gene homology is actually a good case in point. Our short HCR FISH probes that worked gave exactly the same gene-specific signals as the longer RNA probes for the two genes. In other words, despite the high similarity overall (~90%), and shorter stretches of 100% homology, there was no cross reaction by our RNA probes under our hybridization conditions. It remains possible, though, that there are some lncRNA genes canvassed that may have more extensive regions of extremely high identity, due perhaps to duplications, but we did not notice any of these in our fairly extensive examinations of genes analyzed thus far.

3) It is also advised to perhaps perform a non RNA-FISH method to validate the global RNA-FISH patterns presented. For example, RT-qPCR using random hexamers (to avoid the noted poly-A issues in expression measurements) to create cDNA and measure abundance in the male reproductive tissues. This perhaps could be done by use fluorescent markers to isolate the same samples presented and measure abundance. This again could be done with 1-10% of random selected examined lncRNAs to compare to the RNA-FISH results.

We have essentially already done this. For example, HCR-FISH and smFISH analyses duplicate our results with RNA FISH. In our RT-PCR experiments, such as shown in Fig 1h, we have used random hexamer, oligo dT and 3'end-specific primers for reverse transcription. For transcripts that have no PAS motifs, and that were readily detected by FISH vs. RNAseq, RT and subsequent PCR was successful only when using the 3'-specific or random hexamer RT primers. As noted earlier, we see excellent concordance between our FISH results and those of RNAseq for the ~8,000 coding genes that we have analyzed. It is only a subset of the lncRNA genes that consistently do not correlate, and these are enriched for lncRNAs that do not have poly A signals or tails.

While it is expected orthogonal methods may not validate all the results by RNA-FISH it may provide the reader with more assurance that all results are not based on one method. Moreover, a "confidence" level for re-examination of specific lncRNAs presented in this study using orthogonal methodologies. Yet this reviewer understands that there are many lncRNA studies based on one approach (e.g., RNAseq). Thus, it is suggested but perhaps outside the scope to validate the expression patterns reported by at least one or two orthogonal methods -- especially considering the importance of the implications of the findings in this study.

Again, we appreciate the reviewer's concern and generally reasonable suggestions. We feel the responses above should cover these concerns adequately.

4) It would be very interesting if the authors could examine if noncoding RNA is paternally inherited. This would be of great interest to the readership of Nature Communications. Perhaps this implication could be further evidenced by a female KO of

one of these genes and perform RNA-FISH on the maternal abundance of these RNAs upon fertilization.

We agree that this would be a very cool thing to include and had already tried a couple of experiments using the opposite approach of testing unfertilized embryos for loss of signals. The problem with doing this via FISH is that the amount of RNA entering the embryo from the sperm or seminal fluid is likely very low and difficult to detect yet could still be highly effective epigenetically. In the case of the unfertilized eggs, there may also be maternal allele contributions that mask or mirror the paternal ones. We are setting up to do PCR and live tracking-based approaches, but these will take time to set up and conduct.

5) Perhaps the title should focus on expression patterns rather than "roles" which implies function. While it is understandable that the authors correlate their expression findings with genetic studies that truly imply function - yet the title has an initial implication that this is a comprehensive genetic functional screen. Moreover, it may be worth mentioning this study is in *Drosophila* as it may be distinctive from mammalian male reproductive tissues - as many lncRNAs are not conserved as deep as *Drosophila*. This is not to discount the key results in this study but to distinguish evolutionary distances of equal importance.

A common hesitation among the reviewers is that the many of the genetic requirement validations for lncRNAs studied here were not carried out by us. Nevertheless, they have been done (see partial list above) and clearly demonstrate the requirements of many of the genes that we focused on. Taken together with our data and analyses, these analyses support the critical and novel functions indicated here. lncRNAs are also most highly expressed in mammalian testes (as are coding genes due to open chromatin state) where they also play a role in repetitive element management. As we now point out, the human Y chromosome has also recently been shown to contain numerous mega genes, several of which encode lncRNAs. In terms of the title, it has been changed to include *Drosophila* and the term spatial to more accurately describe the content and findings of the study. However, we do make mention of the major roles implied by the combination of our work and the previous work of others, as we believe these are justified and better inform the reader of the study's findings and implications.

Reviewer 3

Shao's paper determined the subcellular localization of a number of lncRNAs in fly male reproductive organs mainly by using in situ hybridization. The imaging quality is good and the work can provide some clues to further functional and mechanism studies of individual lncRNAs. So, it looks fine as a resource paper. However, the research is focused only on a restricted area and lacks deeper studies, thus, it hardly arouses broad interest. I do not think there are enough innovative findings to be published in *Nature Communications*. Please see my specific comments below.

We hope the reviewer will be persuaded otherwise after reading our responses above, along with all of the additions and modifications made in the revised submission. These findings will have a very wide and profound impact.

1. In introduction part, the description of both lncRNA and spermatogenesis is superficial. I could not get much information there. Important lncRNA function and action mechanism, and some critical pathways in spermatogenesis should be described.

We thank the reviewer for pointing out the minimal background content on lncRNAs and spermatogenesis in the introduction. This was due in part to trying to keep our word count down to *Nat Comm* requirements, with much of the descriptions coming later as needed. That said, we have reworked the introduction, first figure and results significantly in order to provide more background information up front. We think this will help the general reader, and we thank the reviewer for the suggestion.

2. Authors proposed many possibilities but could not prove them.

As pointed out in our comments above, the combination of our observations, follow-up analyses and work by others corroborate the functions of many lncRNAs during spermatogenesis. Figuring out the details of these functions will be the domain of hundreds of subsequent projects and researchers. As with our previous resource-styled studies, our new datasets will provide a wealth of ideas and support that help catalyze these ideas and studies.

REVIEWERS' COMMENTS

Reviewer #1 (Remarks to the Author):

The authors have addressed most of the comments. We are however left with one issue as the manuscript still lack functional validation. I do understand that some of these transcripts have been validated by others but I do believe the validation of a newly identified lncRNA will increase the impact of the work. While I do think lncRNAs are extremely important in many biological processes, many researchers are still doubting that they are truly functional so any additional validation will definitely increase the impact of the work.

Reviewer #2 (Remarks to the Author):

The reviewers have present clarification to my concerns. It is clear that this is a resource paper to contribute to the Drosophila community with findings such as many lncRNAs lack PolyA tails and are cytoplasmic. As such the reviewers have addressed my concerns.